# The Effects of a 24-Week Combined Circuit Training and Mobility Program on the Physical Fitness and Body Composition of an Adult Academic Community

**DOI:** 10.3390/sports13030079

**Published:** 2025-03-06

**Authors:** Lorenzo Pugliese, Chiara Tuccella, Gabriele Maisto, Emanuele D’Angelo, Simona Delle Monache, Maria Scatigna, Maria Helena Rodrigues Moreira, Valerio Bonavolontà, Maria Giulia Vinciguerra

**Affiliations:** 1Department of Biotechnological and Applied Clinical Sciences, University of L’Aquila, 67100 L’Aquila, Italy; lorenzo.pugliese@univaq.it (L.P.); chiara.tuccella1@student.univaq.it (C.T.); gabriele.maisto@graduate.univaq.it (G.M.); simona.dellemonache@univaq.it (S.D.M.); mariagiulia.vinciguerra@univaq.it (M.G.V.); 2Department of Neurosciences, Biomedicine and Movement Sciences, University of Verona, 37131 Verona, Italy; 3Department of Wellbeing, Nutrition and Sport, Pegaso Telematic University, 80143 Naples, Italy; emanuele.dangelo@unipegaso.it; 4Department of Life, Health and Environmental Sciences, University of L’Aquila, 67100 L’Aquila, Italy; maria.scatigna@univaq.it; 5Department of Sport Science, Exercise and Health, University of Trás-os-Montes and Alto Douro, 5000-801 Vila Real, Portugal; hmoreira@utad.pt

**Keywords:** physical activity promotion, workplace setting, physical fitness, body composition, circuit training, university, employees

## Abstract

Workplace physical activity programs (WPAPs) are increasingly being recognized for improving employee health, though the results remain inconsistent. Universities provide favorable settings for WPAPs. This study examined the effects of a 24-week circuit training and mobility program on the physical fitness (PF) and body composition of university employees. Thirty-eight university employees (9 males and 29 females; 51.5 ± 12.6 years) followed a 24-week training program consisting of two circuit training sessions and one mobility session per week. PF (including cardiorespiratory fitness (CRF), grip strength, shoulder mobility, core endurance, lower limb strength, and balance) body composition, anthropometric variables, and physical activity (PA) levels were assessed at baseline (T0), 12 weeks (T1), and 24 weeks (T2). CRF, strength, mobility, and core endurance significantly improved. A body composition analysis indicated a decrease in fat mass and an increase in lean mass at T2. Minor changes were observed in anthropometric variables. Furthermore, PA levels increased throughout the intervention. The 24-week WPAP improved the PF and body composition of university employees.

## 1. Introduction

The promotion of physical activity (PA) in different contexts is a strategical asset for the enhancement of health status and the implementation of healthy habits across different populations [1].

Physical inactivity (PI) and sedentary behavior (SB) are established risk factors for non-communicable diseases (NCDs), with recent evidence highlighting their contribution to disease onset [2,3].

Despite substantial evidence supporting the positive effects of PA on NCD prevention, the global rates of insufficient PA continue to rise, increasing from 23.4% in 2000 to 31.3% in 2022 [4]. The World Health Organization (WHO) has addressed this concern through the Global Action Plan of Physical Activity 2018–2030, emphasizing workplace-based interventions as a key strategy [1].

The workplace environment presents a significant challenge, with employees spending approximately 70–85% of their working hours performing sedentary activities [5,6]. Research indicates that occupational sitting time exceeds non-occupational sedentary behavior, creating a substantial barrier to PA participation [7]. Consequently, the workplace has emerged as a critical setting for implementing PA interventions aimed at improving employee health outcomes [8,9,10,11,12]. For these very reasons, the University of L’Aquila has devised a workplace physical activity program called “Ateneo in Movimento” (“University on move”) with the aim of promoting active lifestyles in the university environment.

Circuit training (CT) can be considered particularly promising as a workplace physical activity program (WPAP) strategy. This training modality involves performing sequential exercises at moderate to vigorous intensity with minimal rest intervals, providing a time-efficient approach to exercise [13]. Previous studies have documented CT’s effectiveness in improving cardiorespiratory fitness (CRF), physical function, body composition, and anaerobic performance [14,15]. Furthermore, when compared to traditional walking programs, CT has demonstrated superior outcomes in improving body composition parameters and muscle mass [5].

The combination of CT with mobility sessions is particularly relevant in workplace settings, where prolonged sitting has been associated with reduced flexibility and postural alterations [16,17,18]. Workplace-based flexibility programs have shown effectiveness in reducing musculoskeletal discomfort, improving work-related musculoskeletal symptoms and range of motion, particularly in office workers [19].

University settings are ideal for WPAPs due to their built-in facilities, flexible schedules, and health-conscious populations. Higher education institutions provide structured environments where interventions can be systematically implemented and evaluated [20]. WPAPs have demonstrated multiple benefits, including improved sleep quality, reduced stress and anxiety levels, enhanced job satisfaction, and increased workplace productivity [21]. Additionally, these programs have shown positive effects on workplace collaboration and employee well-being [22].

A combined CT and mobility intervention appear particularly suited to the university environment due to its adaptability and time efficiency. However, previous research has identified lack of time as a primary barrier to workplace exercise participation [22], while program variety has been recognized as a key factor in maintaining engagement [23]. The flexible nature of a combined CT and postural program allows for adaptation to varying fitness levels and schedules, making it potentially ideal for implementation in academic settings. To the best of our knowledge, studies that used CT and mobility sessions as interventions to promote PA in adult university employees are lacking.

Therefore, this study aimed to evaluate the effects of a 24-week combined CT and mobility intervention program on physical fitness and body composition in university employees. Based on previous evidence, we hypothesized that participants would show improvements in CRF, muscular strength, flexibility, and body composition measures.

## 2. Materials and Methods

### 2.1. Study Design

This longitudinal intervention study evaluated the effects of a 24-week CT and mobility program. Measurements were taken at baseline (T0), after 12 weeks (T1), and after 24 weeks (T2).

### 2.2. Participants

Thirty-eight university employees (9 males and 29 females; age: 51.5 ± 12.6 years; body height: 166.7 ± 9.0 cm; body mass: 69.8 ± 11.9 kg) volunteered to participate in the study. Inclusion criteria were (1) full-time employment at the University of L’Aquila, (2) an age between 25 and 65 years, and (3) medical clearance for PA. Exclusion criteria included (1) cardiovascular, respiratory, or metabolic diseases and (2) musculoskeletal conditions limiting exercise participation. The study was conducted in accordance with the Declaration of Helsinki and approved by the institutional review board (approval number: 17/2020). All participants provided written informed consent. 

### 2.3. Training Protocol

The training intervention consisted of three supervised 45 min sessions per week over a 24-week period. The participants performed two training sessions per week of a circuit training program and one training session per week of mobility exercises. Sessions began with a comprehensive 10 min warm-up protocol incorporating dynamic stretching sequences and progressive aerobic activities to prepare participants for the main workout phase.

The main circuit training component (Appendix A) lasted 24 min and was developed following the fitness intervention principles for healthy adults as outlined by Garber et al. [24]. The circuit included eight stations following a work–rest interval of 30 s of exercise followed by 30 s of rest, alternating between resistance and aerobic exercises to support both cardiorespiratory and muscular adaptations [25] (Appendix A). Exercises included multi-joint movements, with aerobic intervals integrated between resistance exercises. 

The mobility session involved whole-body yoga and Pilates exercises, which incorporated sequences from various positions (e.g., upright, supine, prone, and lying on the side) using body weight and elastic bands. Each exercise lasted 15–30 s or for 12–20 repetitions with 2–4 sets as needed (Appendix A). 

Exercise intensity was monitored using the Borg Rating of Perceived Exertion scale of 6–20 [26], maintaining a target range of 13–15, which corresponds to the recommended intensity for improving cardiorespiratory fitness in sedentary adults [27]. To ensure continued adaptation and maintain training effectiveness, progressive overload was implemented systematically every four weeks. This included planned progressions in repetitions, resistance, and exercise complexity based on individual participant progress [28]. 

Each session concluded with a 10 min cool-down period, incorporating static stretching and relaxation techniques. All training sessions were supervised by experts in sports science.

### 2.4. Outcome Measures

A comprehensive battery of assessments was conducted to evaluate multiple components of physical fitness following standardized protocols. CRF was assessed using the 2 min step test validated as a measure of cardiorespiratory endurance in adult populations [28]. Handgrip strength was measured using a calibrated Saehan dynamometer following the standardized protocol recommended by Roberts et al. [29]. Core endurance was assessed via the standardized sit-up test [30]. Lower body power was measured using the jump and reach test following standardized protocols [31]. The shoulder neck mobility test was used to assess upper body flexibility [32]. The one-leg stand test was performed to evaluate balance following established protocols [33].

Anthropometric measurements were conducted following standardized procedures recommended by the International Society for the Advancement of Kinanthropometry [34]. This included height, weight, BMI calculation, and circumference measurements of the waist, hip, thigh, and arm. Body composition (fat mass and fat-free mass) was assessed using bioelectrical impedance analysis with the Wunder mod. WBA300 (Wunder SA.BI.SRL, Trezzo sull’Adda, Milan, Italy).

Physical activity levels were assessed using the International Physical Activity Questionnaire short form (IPAQ-SF), validated across multiple populations [35,36].

### 2.5. Statistical Analysis

Data were analyzed using one-way repeated measures ANOVA to examine changes over time (IBM SPSS statistics, version 29). Mauchly’s test assessed sphericity, with Greenhouse–Geisser corrections applied when necessary. Post hoc analyses used Bonferroni corrections for multiple comparisons. Effect sizes were calculated using partial eta-squared (η^2^*p*) and Cohen’s d. Statistical significance was set at *p* < 0.05.

## 3. Results

The 24-week CT and mobility program yielded significant improvements across multiple fitness parameters, with all 38 participants completing the intervention. 

As shown in Table 1, cardiorespiratory fitness demonstrated substantial improvement throughout the intervention period (*p* = 0.01, η^2^p = 0.221). Post hoc analyses revealed significant increases at every time point, with notable progress occurring during the first 12 weeks and continued enhancement seen through the program’s completion (Appendix A).

Muscular strength assessments indicated consistent improvements across multiple measures (Table 1). Handgrip strength (*p* = 0.01, η^2^p = 0.225) and the jump and reach test (*p* = 0.019, η^2^p = 0.198) showed significant enhancement over time, with statistically significant differences between the baseline and the end of the training program. Core endurance measured via sit-up performance demonstrated significant enhancement between all time points (*p* = 0.01, η^2^p = 0.226).

The body composition analysis revealed modest changes in both fat mass and lean mass. Fat mass showed a trend toward reduction, decreasing from 32.62 ± 8.22% at baseline to 32.56 ± 8.42% and 32.35 ± 8.5% at mid-intervention and at final assessment, respectively (F[2, 74] = 1.060, *p* = 0.357, η^2^p = 0.056). Lean mass demonstrated a marginal increase over the intervention period, with values of 46.9 ± 10.40 kg at baseline, 47.40 ± 10.67 kg at mid-intervention, and 47.02 ± 10.24 kg at completion (F[2, 74] = 2.708, *p* = 0.081, η^2^p = 0.068). 

Anthropometric measurements showed varying responses to the intervention. Thigh circumference showed no significant increases over time (F[2, 74] = 1.419, *p* = 0.241, η^2^p = 0.037), progressing from 52.29 ± 4.78 cm to 53.08 ± 4.30 cm. Waist circumference values were 84.39 ± 11.65 cm at baseline, 83.93 ± 11.35 cm at mid-intervention, and 84.1 ± 11.58 cm at completion (F[2, 74] = 0.551, *p* = 0.555, η^2^p = 0.015). The waist-to-hip ratio remained constant at 0.81 ± 0.08 across all time points (F[2, 74] = 0.267 *p* = 0.716, η^2^p = 0.007).

Physical activity levels, as measured using the IPAQ, demonstrated significant increases throughout the intervention (F[2, 74] = 14.228, *p* < 0.001, η^2^p = 0.371). The values increased from 1444.15 ± 198.39 MET-min/week at baseline to 1587.15 ± 193.94 MET-min/week at mid-intervention before slightly decreasing to 1571.60 ± 154.01 MET-min/week at the final assessment. The magnitude of these changes varied across time points, as detailed in Table 2. No significant differences were found for the one-leg stand test (F[2, 74] = 1.448, *p* = 0.248, η^2^p = 0.074).

## 4. Discussion

Our findings demonstrate that a 24-week WPAP can effectively improve various components of physical fitness among university employees. The most substantial improvements were observed in cardiorespiratory fitness, muscular strength, flexibility, and PA levels. These results are particularly significant given the sedentary nature of academic work and the growing recognition of workplace health promotion as a public health priority [37].

The significant improvement in the 2 min step test performance reflects meaningful cardiorespiratory adaptation. Proper et al. [37], in their comprehensive systematic review of WPAPs, found strong evidence of improved CRF following structured interventions. In our study, participants showed particularly strong improvements during the first 12 weeks, consistent with established patterns of physiological adaptation to structured exercise programs [24]. The continued, albeit slower, improvement in the latter 12 weeks suggests ongoing physiological adaptations, highlighting the importance of maintaining long-term exercise programs. Similar findings were reported by Pedersen et al. [8], who demonstrated that a workplace intervention can improve CRF and overall health status in participants. Our results align with this observation, supporting the effectiveness of WPAPs for improving cardiovascular fitness.

The increases in handgrip strength, shoulder mobility, and sit-up performance demonstrate the effective enhancement of both upper body strength and mobility and core endurance. These results are consistent with findings from other studies. Ramos-Campo et al. [13] found improvements in strength parameters in their meta-analysis, reporting significant pre–post increases in upper limb and lower limb strength following circuit training interventions. Lehnert et al. [15] reported significant improvements in strength endurance parameters following circuit training interventions, particularly noting a significant increase in core strength as measured using the partial curl-up test. Their findings align with our results showing enhanced sit-up performance, suggesting that circuit-based training programs effectively target core musculature, which is essential for maintaining proper posture and reducing work-related musculoskeletal discomfort in office settings. Karatrantou et al. [19] similarly found that a 12-week combined circuit strength training and mobility program effectively improved flexibility and strength endurance while reducing musculoskeletal pain in office employees. The improvements observed in our study are particularly relevant for office workers, who typically experience decreased muscular strength and increased musculoskeletal complaints due to prolonged sitting and computer use. Eerd et al. [38] identified strength training as a key intervention for preventing work-related musculoskeletal disorders, with moderate evidence supporting its effectiveness in reducing upper extremity complaints. Furthermore, Buckley et al. [39] emphasized that improved muscular strength and endurance can enhance work capacity and reduce the risk of work-related injuries in office environments.

The significant improvements in PA levels are particularly noteworthy as they indicate potential behavioral changes extending beyond the supervised exercise sessions. Watanabe and Kawakami [10] reported similar findings in their multi-component workplace intervention, showing significant increases in overall physical activity levels among employees in intervention worksites compared to control sites. They emphasized that workplace environments with employer support, incentives, and exercise programs can enhance employees’ awareness of physical activity importance and reduce barriers to behavioral change. Abraham and Graham-Rowe’s [40] meta-analysis of workplace interventions found that structured programs can lead to sustained increases in physical activity, reporting a significant positive effect (d = 0.29, 95% CI [0.15, 0.43]). This finding suggests that WPAPs may serve as behavioral change catalysts, potentially through improved self-efficacy and enhanced awareness of exercise benefits. The development of regular exercise habits within the workplace setting may facilitate the transfer of these behaviors to other life contexts [41].

The lack of significant changes in anthropometric measures suggests that exercise interventions alone might have a limited impact on body composition. This finding aligns with previous research indicating that combined exercise and dietary interventions might be more effective in producing changes in body composition parameters [37]. The effectiveness of our program may be attributed to several key design elements identified in previous research. First, the combination of different training modalities (cardiovascular, strength, and flexibility) addresses multiple fitness components, which Garber et al. [24] identified as crucial for comprehensive health benefits. Additionally, the workplace setting helps eliminate common barriers to exercise participation, such as travel time and facility access, which often hinder engagement in physical activity [39]. The multi-component nature of our intervention appears to have contributed to its effectiveness, similar to previous research demonstrating that flexible, multi-component workplace programs produced significant improvements in physical activity levels [10,19]. Grimani et al. [9] suggested that workplace interventions addressing multiple levels (individual and organizational) show greater promise for improving health outcomes and work-related parameters.

### Limitations and Future Research 

The study limitations include some methodological considerations. First, the single-group design and potential selection bias, as participants volunteered for the program, as well as low gender equality, may affect the internal validity of our findings. These limitations, while common in workplace intervention studies, necessitate careful interpretation of the results. Another limitation is that PA levels outside of the structured training sessions and adherence to the WPAP were not measured or controlled for. This lack of control over external physical activity makes it difficult to attribute the observed improvements solely to the CT program as participants may have engaged in additional exercise or modified their activity patterns outside of the intervention.

Future studies should incorporate objective measures of PA throughout the intervention period to better understand the total PA dose and its relationship with the observed outcomes. Moreover, as suggested in previous research [15], a decrease in rest periods during CT across the intervention period could contribute to more impactful training stimuli. Additionally, the lack of a dietary component limits our understanding of the program’s comprehensive health impact.

## 5. Conclusions

The 24-week “Ateneo in Movimento” CT and mobility program effectively improved PF parameters among university employees, particularly in CRF and muscular strength. These improvements, consistent with previous research, demonstrate the potential of structured WPAPs in academic settings. The observed generalization of effects on general PA levels suggests broader behavioral impacts beyond the immediate program outcomes. Future research should incorporate randomized controlled designs, dietary components, and longer follow-up periods to better understand the comprehensive and sustained effects of WPAPs. Additionally, investigating the cost-effectiveness and implementation challenges of such programs in various workplace settings would provide valuable insights for public health policy and practices. 

## Figures and Tables

**Table 1 sports-13-00079-t001:** Physical performance measures across time points.

	T0	T1	T2	F(2, 74)	*p*-Value	η^2^p
2 min Step in Place (n)	84.65 ± 19.02	90.73 ± 17.96 *	91.81 ± 18.69 *^§^	5.104	0.01	0.221
Handgrip (kg)	29.73 ± 8.02	31.02 ± 8.28	31.73 ± 8.20 *	5.234	0.01	0.225
Sit-Up (n)	12.42 ± 4.48	13.26 ± 4.01 *	13.42 ± 3.69 *^§^	5.241	0.01	0.226
Jump and Reach (cm)	23.48 ± 9.96	24.88 ± 9.88	25.80 ± 9.74 *	4.431	0.019	0.198
Shoulder Neck Test (dx) (n)	3.50 ± 1.35	3.57 ± 1.13	4.07 ± 1.09 *	8.355	0.001	0.184
Shoulder Neck Test (sx) (n)	3.39 ± 1.46	3.50 ± 1.33	3.92 ± 1.36 *	3.461	0.042	0.161

Values are presented as mean ± SD; η^2^p = partial eta squared. Effect sizes: small (η^2^p ≥ 0.01), medium (η^2^p ≥ 0.06), large (η^2^p ≥ 0.14). *: significantly different from T0; ^§^: significantly different from T1.

**Table 2 sports-13-00079-t002:** Effect sizes (Cohen’s d) for significant changes.

	T0–T1	T1–T2	T0–T2
2 min Step in Place	0.33	0.05	0.37
Handgrip	0.16	0.09	0.25
Sit-Up	0.21	0.04	0.24
Jump and Reach	0.14	0.09	0.24
Shoulder Neck Test (dx)	0.06	0.45	0.46
Shoulder Neck Test (sx)	0.08	0.38	0.37
IPAQ-SF	0.73	−0.09	0.71

Effect sizes were interpreted as follows: d < 0.2 indicates a negligible effect, 0.2 ≤ d < 0.5 indicates a small effect, 0.5 ≤ d < 0.8 indicates a medium effect, and d ≥ 0.8 indicates a large effect.

## Data Availability

The data will be made available upon reasonable request.

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
