# Peer review of "The Effects of a 24-Week Combined Circuit Training and Mobility Program on the Physical Fitness and Body Composition of an Adult Academic Community"

_sports, 2025, doi:10.3390/sports13030079_

Round 1

Reviewer 1 Report

Comments and Suggestions for Authors

The article „Effects of 24-week combined Circuit Training and Mobility  Program on Physical Fitness and Body Composition of an  Adult Academic Community „ is well-written  and ovel and deals with important aspects of maintaining health. 

The abstract is well-written and provides a good summary of the results. However, it should not include the BMI results, which do provide much unclear information than body composition. Generally, since the authors provided full body composition measures, the reporting BMI does not make much sense since increased lean mass cause typically BMI to increase. Thus BMI might be briefly mentioned in the discussion but not the main results in the abstract. Anyway, I suggest removing BMI from the statistical analyses. The abstract in sstem is too long.
Although the manuscript includes good relevant data, they are purely presented. Therefore, I highly recommend adding a graphical presentation of significant results, best with individual data visualization. So far, we don’t know whether all participants improved or whether some of them might stagnate in improvement between T1 and T2.
Please add the supplementary material with the full scheme and exercise figures, and system of your intervention. This is the most important part of possible reproducibility. 
The limitation section should include low gender equality, lack of movement outside the university, and adherence monitoring. Monitoring Physical activity levels by IPAQ is not a direct measurement. 

Line 57-61: Does circuit training improve anaerobic performance?:
Lehnert, M., Stastny, P., Sigmund, M., Xaverova, Z., Hubnerova, B., & Kostrzewa, M. (2015). The effect of combined machine and body weight circuit training on women's muscle strength and body composition. Journal of Physical Education and Sport, 15(3), 561.

Line 83? please summarize the research gaps before stinging the study aim.

Discussion is fair, but it should include the direct comparison to previous circuit trainings.

Reviewer 2 Report

Comments and Suggestions for Authors

This study evaluated the impact of a 24-week workplace physical activity programme on the physical fitness and body composition of university employees. The results demonstrated significant improvements in cardiorespiratory fitness, upper limb strength, and core endurance, accompanied by a decrease in fat mass and an increase in lean mass. The intervention was found to be effective in enhancing the physical fitness and body composition of the participants.

It is important to acknowledge the contribution of the authors to the enhancement of the physical fitness and well-being of workers, with a particular emphasis on those who engage in sedentary work.

Is the workplace exercise programme called 'Ateneo in Movimento' the training protocol described? Please provide details

Introduction: In addition to the abstract, it is requested that a description of the other parameters and their abbreviations be included in the main text, as has been done for 'Physical Activity (PA)'.

Results: The presentation of tables (as Table 1) accompanied by the other parameters would be advantageous. Furthermore, the tables should elucidate the significant differences between the intervention periods with greater clarity.

Discussion: Dear authors, the article is well-written, yet the discussion is lacking. It would be beneficial if you could strengthen it. 

Round 2

Reviewer 1 Report

Comments and Suggestions for Authors

I´m satisfied with the answers. There is just one minor issue that the description of supplementary materials should be more detailed and explicit. Moreover, some individual dt would still help. Otherwise the manuscript is worth to be published.